# Collaborative PAC Learning

**Avrim Blum**
Toyota Technological Institute at Chicago
Chicago, IL 60637
avrim@ttic.edu

**Nika Haghtalab**
Computer Science Department
Carnegie Mellon University
Pittsburgh, PA 15213
nhaghtal@cs.cmu.edu

**Ariel D. Procaccia**
Computer Science Department
Carnegie Mellon University
Pittsburgh, PA 15213
arielpro@cs.cmu.edu

**Mingda Qiao**
Institute for Interdisciplinary Information Sciences
Tsinghua University
Beijing, China 100084
qmd14@mails.tsinghua.edu.cn

## Abstract

We consider a collaborative PAC learning model, in which $k$ players attempt to learn the same underlying concept. We ask how much more information is required to learn an accurate classifier for all players simultaneously. We refer to the ratio between the sample complexity of collaborative PAC learning and its non-collaborative (single-player) counterpart as the *overhead*. We design learning algorithms with $O(\ln(k))$ and $O(\ln^2(k))$ overhead in the *personalized* and *centralized* variants our model. This gives an exponential improvement upon the naïve algorithm that does not share information among players. We complement our upper bounds with an $\Omega(\ln(k))$ overhead lower bound, showing that our results are tight up to a logarithmic factor.

## 1 Introduction

According to Wikipedia, *collaborative learning* is a "situation in which two or more people learn ... something together," e.g., by "capitalizing on one another's resources" and "asking one another for information." Indeed, it seems self-evident that collaboration, and the sharing of information, can make learning more efficient. Our goal is to formalize this intuition and study its implications.

As an example, suppose $k$ branches of a department store, which have sales data for different items in different locations, wish to collaborate on learning which items should be sold at each location. In this case, we would like to use the sales information across different branches to learn a good policy for each branch. Another example is given by $k$ hospitals with different patient demographics, e.g., in terms of racial or socio-economic factors, which want to predict occurrence of a disease in patients. In addition to requiring a classifier that performs well on the population served by each hospital, it is natural to assume that all hospitals deploy a common classifier.

Motivated by these examples, we consider a model of *collaborative PAC learning*, in which $k$ players attempt to learn the same underlying concept. We then ask how much information is needed for all players to simultaneously succeed in learning a desirable classifier. Specifically, we focus on the classic *probably approximately correct (PAC)* setting of Valiant [14], where there is an unknown target function $f^* \in \mathcal{F}$. We consider $k$ players with distributions $D_1, \ldots, D_k$ that are labeled according to $f^*$. Our goal is to learn $f^*$ up to an error of $\epsilon$ on *each and every* player distribution while requiring only a small number of samples overall.

A natural but naïve algorithm that forgoes collaboration between players can achieve our objective by taking, from each player distribution, a number of samples that is sufficient for learning the individual task, and then training a classifier over all samples. Such an algorithm uses $k$ times as many samples as needed for learning an individual task — we say that this algorithm incurs $O(k)$ *overhead* in sample complexity. By contrast, we are interested in algorithms that take advantage of the collaborative environment, learn $k$ tasks by sharing information, and incur $o(k)$ overhead in sample complexity.

We study two variants of the aforementioned model: *personalized* and *centralized*. In the personalized setting (as in the department store example), we allow the learning algorithm to return different functions for different players. That is, our goal is to return classifiers $f_1, \ldots, f_k$ that have error of at most $\epsilon$ on player distributions $D_1, \ldots, D_k$, respectively. In the centralized setting (as in the hospital example), the learning algorithm is required to return a *single* classifier $f$ that has an error of at most $\epsilon$ on all player distributions $D_1, \ldots, D_k$. Our results provide upper and lower bounds on the sample complexity overhead required for learning in both settings.

## 1.1 Overview of Results

In Section 3, we provide algorithms for personalized and centralized collaborative learning that obtain exponential improvements over the sample complexity of the naïve approach. In Theorem 3.1, we introduce an algorithm for the personalized setting that has $O(\ln(k))$ overhead in sample complexity. For the centralized setting, in Theorem 3.2, we develop an algorithm that has $O(\ln^2(k))$ overhead in sample complexity. At a high level, the latter algorithm first learns a series of functions on adaptively chosen mixtures of player distributions. These mixtures are chosen such that for any player a large majority of the functions perform well. This allows us to combine all functions into one classifier that performs well on every player distribution. Our algorithm is an improper learning algorithm, as the combination of these functions may not belong to $\mathcal{F}$.

In Section 4, we present lower bounds on the sample complexity of collaborative PAC learning for the personalized and centralized variants. In particular, in Theorem 4.1 we show that any algorithm that learns in the collaborative setting requires $\Omega(\ln(k))$ overhead in sample complexity. This shows that our upper bound for the personalized setting, as stated in Theorem 3.1, is tight. Furthermore, in Theorem 4.5, we show that obtaining *uniform convergence* across $\mathcal{F}$ over all $k$ player distributions requires $\Omega(k)$ overhead in sample complexity. Interestingly, our centralized algorithm (Theorem 3.2) bypasses this lower bound by using arguments that do not depend on uniform convergence. Indeed, this can be seen from the fact that it is an improper learning algorithm.

In Appendix D, we discuss the extension of our results to the non-realizable setting. Specifically, we consider a setting where there is a "good" but not "perfect" target function $f^* \in \mathcal{F}$ that has a small error with respect to every player distribution, and prove that our upper bounds carry over.

## 1.2 Related Work

Related work in computational and statistical learning has examined some aspects of the general problem of learning multiple related tasks simultaneously. Below we discuss papers on multi-task learning [4, 3, 7, 5, 10, 13], domain adaptation [11, 12, 6], and distributed learning [2, 8, 15], which are most closely related.

Multi-task learning considers the problem of learning multiple tasks in series or in parallel. In this space, Baxter [4] studied the problem of model selection for learning multiple related tasks. In their work, each learning task is itself randomly drawn from a distribution over related tasks, and the learner's goal is to find a hypothesis space that is appropriate for learning all tasks. Ben-David and Schuller [5] also studied the sample complexity of learning multiple related tasks. However, in their work similarity between two tasks is represented by existence of "transfer" functions though which underlying distributions are related.

Mansour et al. [11, 12] consider a multi-source domain adaptation problem, where the learner is given $k$ distributions and $k$ corresponding predictors that have error at most $\epsilon$ on individual distributions. The goal of the learner is to combine these predictors to obtain error of $k\epsilon$ on any unknown mixture of player distributions. Our work is incomparable to this line of work, as our goal is to learn classifiers, rather than combining existing ones, and our benchmark is to obtain error $\epsilon$ on each individual distribution. Indeed, in our setting one can learn a hypothesis that has error $k\epsilon$ on any mixture of players with no overhead in sample complexity.

Distributed learning [2, 8, 15] also considers the problem of learning from $k$ different distributions simultaneously. However, the main objective in this space is to learn with limited communication between the players, rather than with low sample complexity.

## 2 Model

Let $\mathcal{X}$ be an instance space and $\mathcal{Y} = \{0, 1\}$ be the set of labels. A *hypothesis* is a function $f : \mathcal{X} \to \mathcal{Y}$ that maps any instance $x \in \mathcal{X}$ to a label $y \in \mathcal{Y}$. We consider a *hypothesis class* $\mathcal{F}$ with VC dimension $d$. Given a distribution $D$ over $\mathcal{X} \times \mathcal{Y}$, the *error* of a hypothesis $f$ is defined as $\mathrm{err}_D(f) = \Pr_{(x,y) \sim D}[f(x) \neq y]$.

In the *collaborative learning* setting, we consider $k$ players with distributions $D_1, \ldots, D_k$ over $\mathcal{X} \times \mathcal{Y}$. We focus on the *realizable* setting, where all players' distributions are labeled according to a common target function $f^* \in \mathcal{F}$, i.e., $\mathrm{err}_{D_i}(f^*) = 0$ for all $i \in [k]$ (but see Appendix D for an extension to the non-realizable setting). We represent an instance of the collaborative PAC learning setting with the 3-tuple $(\mathcal{F}, f^*, \{D\}_{i \in [k]})$.

Our goal is to learn a good classifier with respect to *every* player distribution. We call this $(\epsilon, \delta)$-*learning* in the collaborative PAC setting, and study two variants: the *personalized* setting, and the *centralized* setting. In the personalized setting, our goal is to learn functions $f_1, \ldots, f_k$, such that with probability $1 - \delta$, $\mathrm{err}_{D_i}(f_i) \leq \epsilon$ for all $i \in [k]$. In the centralized setting, we require all the output functions to be identical. Put another way, our goal is to return a single $f$, such that with probability $1 - \delta$, $\mathrm{err}_{D_i}(f) \leq \epsilon$ for all $i \in [k]$. In both settings, we allow our algorithm to be *improper*, that is, the learned functions need not belong to $\mathcal{F}$.

We compare the sample complexity of our algorithms to their PAC counterparts in the realizable setting. In the traditional realizable PAC setting, $m_{\epsilon, \delta}$ denotes the number of samples needed for $(\epsilon, \delta)$-learning $\mathcal{F}$. That is, $m_{\epsilon, \delta}$ is the total number of samples drawn from a realizable distribution $D$, such that, with probability $1 - \delta$, any classifier $f \in \mathcal{F}$ that is consistent with the sample set satisfies $\mathrm{err}_D(f) \leq \epsilon$. We denote by $\mathcal{O}_{\mathcal{F}}(\cdot)$ the function that, for any set $S$ of labeled samples, returns a function $f \in \mathcal{F}$ that is consistent with $S$ if such a function exists (and outputs "none" otherwise). It is well-known that sampling a set $S$ of size $m_{\epsilon, \delta} = O\left(\frac{1}{\epsilon}\left(d \ln\left(\frac{1}{\epsilon}\right) + \ln\left(\frac{1}{\delta}\right)\right)\right)$, and applying $\mathcal{O}_{\mathcal{F}}(S)$, is sufficient for $(\epsilon, \delta)$-learning a hypothesis class $\mathcal{F}$ of VC dimension $d$ [1]. We refer to the ratio of the sample complexity of an algorithm in the collaborative PAC setting to that of the (non-collaborative) realizable PAC setting as the *overhead*. For ease of exposition, we only consider the dependence of the overhead on parameters $k$, $d$, and $\epsilon$.

## 3 Sample Complexity Upper Bounds

In this section, we prove upper bounds on the sample complexity of $(\epsilon, \delta)$-learning in the collaborative PAC setting. We begin by providing a simple algorithm with $O(\ln(k))$ overhead (in terms of sample complexity, see Section 2) for the personalized setting. We then design and analyze an algorithm for the centralized setting with $O(\ln^2(k))$ overhead, following a discussion of additional challenges that arise in this setting.

### 3.1 Personalized Setting

The idea underlying the algorithm for the personalized setting is quite intuitive: If we were to learn a classifier that is on average good for the players, then we have learned a classifier that is good for a large fraction of the players. Therefore, a large fraction of the players can be simultaneously satisfied by a single good global classifier. This process can be repeated until each player receives a good classifier.

In more detail, let us consider an algorithm that pools together a sample set of total size $m_{\epsilon/4, \delta}$ from the uniform mixture $D = \frac{1}{k} \sum_{i \in [k]} D_i$ over individual player distributions, and finds $f \in \mathcal{F}$ that is consistent with this set. Clearly, with probability $1 - \delta$, $f$ has a small error of $\epsilon/4$ with respect to distribution $D$. However, we would like to understand how well $f$ performs on each individual player's distribution.

Since $\mathrm{err}_D(f) \leq \epsilon/4$ is also the *average error* of $f$ on player distributions, with probability $1 - \delta$, $f$ must have error of at most $\epsilon/2$ on at least half of the players. Indeed, one can identify such players by taking additional $\tilde{O}(\frac{1}{\epsilon})$ samples from each player and asking whether the empirical error of $f$ on these sample sets is at most $3\epsilon/4$. Using a variant of the VC theorem, it is not hard to see that

for any player $i$ such that $\mathrm{err}_{D_i}(f) \leq \epsilon/2$, the empirical error of $f$ is at most $3\epsilon/4$, and no player with empirical error at most $3\epsilon/4$ has true error that is worst than $\epsilon$. Once players with empirical error $3\epsilon/4$ are identified, one can output $f_i = f$ for any such player, and repeat the procedure for the remaining players. After $\log(k)$ rounds, this process terminates with all players having received functions with error of at most $\epsilon$ on their respective distributions, with probability $1 - \log(k)\delta$.

We formalize the above discussion via Algorithm 1 and Theorem 3.1. For completeness, a more rigorous proof of the theorem is given in Appendix A.

---

**Algorithm 1** PERSONALIZED LEARNING

---
$N_1 \leftarrow [k]$; $\delta' \leftarrow \delta/2\log(k)$;
**for** $r = 1, \ldots, \lceil \log(k) \rceil$ **do**
$\quad$ $\tilde{D}_r \leftarrow \frac{1}{|N_r|} \sum_{i \in N_r} D_i$;
$\quad$ Let $S$ be a sample of size $m_{\epsilon/4, \delta'}$ drawn from $\tilde{D}_r$, and $f^{(r)} \leftarrow \mathcal{O}_{\mathcal{F}}(S)$;
$\quad$ Let $G_r \leftarrow \mathrm{TEST}(f^{(r)}, N_r, \epsilon, \delta')$;
$\quad$ $N_{r+1} \leftarrow N_r \setminus G_r$;
$\quad$ **for** $i \in G_r$ **do** $f_i \leftarrow f^{(r)}$;
**end**
**return** $f_1, \ldots, f_k$

---
$\mathbf{TEST}(f, N, \epsilon, \delta)$:

---
**for** $i \in N$ **do** take sample set $T_i$ of size $O\left(\frac{1}{\epsilon} \ln\left(\frac{|N|}{\epsilon\delta}\right)\right)$ from $D_i$ ;
**return** $\{i \mid \mathrm{err}_{T_i}(f) \leq \frac{3}{4}\epsilon\}$

---

**Theorem 3.1.** For any $\epsilon, \delta > 0$, and hypothesis class $\mathcal{F}$ of VC dimension $d$, Algorithm 1 $(\epsilon, \delta)$-learns $\mathcal{F}$ in the personalized collaborative PAC setting using $m$ samples, where

$$ m = O\left(\frac{\ln(k)}{\epsilon}\left((d+k)\ln\left(\frac{1}{\epsilon}\right) + k\ln\left(\frac{k}{\delta}\right)\right)\right). $$

Note that Algorithm 1 has $O(\ln(k))$ overhead when $k = O(d)$.

### 3.2 Centralized Setting

We next present a learning algorithm with $O(\ln^2(k))$ overhead in the centralized setting. Recall that our goal is to learn a *single* function $f$ that has an error of $\epsilon$ on every player distribution, as opposed to the personalized setting where players can receive different functions.

A natural first attempt at learning in the centralized setting is to combine the classifiers $f_1, \ldots, f_k$ that we learned in the personalized setting (Algorithm 1), say, through a weighted majority vote. One challenge with this approach is that, in general, it is possible that many of the functions $f_j$ perform poorly on the distribution of a different player $i$. The reason is that when Algorithm 1 finds a suitable $f^{(r)}$ for players in $G_r$, it completely removes them from consideration for future rounds; subsequent functions may perform poorly with respect to the distributions associated with those players. Therefore, this approach may lead to a global classifier with large error on some player distributions.

To overcome this problem, we instead design an algorithm that continues to take additional samples from players for whom we have already found suitable classifiers. The key idea behind the centralized learning algorithm is to group the players at every round based on how many functions learned so far have large error rates on those players' distributions, and to learn from data sampled from all the groups simultaneously. This ensures that the function learned in each round performs well on a large fraction of the players in *each* group, thereby reducing the likelihood that in later stages of this process a player appears in a group for which a large fraction of the functions perform poorly.

In more detail, our algorithm learns $t = \Theta(\ln(k))$ classifiers $f^{(1)}, f^{(2)}, \ldots, f^{(t)}$, such that for any player $i \in [k]$, at least $0.6t$ functions among them achieve an error below $\epsilon' = \epsilon/6$ on $D_i$. The algorithm then returns the classifier $\mathrm{maj}(\{f^{(r)}\}_{r=1}^t)$, where, for a set of hypotheses $F$, $\mathrm{maj}(F)$ denotes the classifier that, given $x \in \mathcal{X}$, returns the label that the majority of hypotheses in $F$ assign to $x$. Note that any instance that is mislabeled by this classifier must be mislabeled by at least $0.1t$

functions among the $0.6t$ good functions, i.e., $1/6$ of the good functions. Hence, $\text{maj}(\{f^{(r)}\}_{r=1}^t)$ has an error of at most $6\epsilon' = \epsilon$ on each distribution $D_i$.

Throughout the algorithm, we keep track of counters $\alpha_i^{(r)}$ for any round $r \in [t]$ and player $i \in [k]$, which, roughly speaking, record the number of classifiers among $f^{(1)}, f^{(2)}, \ldots, f^{(r)}$ that have an error of more than $\epsilon'$ on distribution $D_i$. To learn $f^{(r+1)}$, we first group distributions $D_1, \ldots, D_k$ based on the values of $\alpha_i^{(r)}$, draw about $m_{\epsilon', \delta}$ samples from the mixture of the distributions in each group, and return a function $f^{(r+1)}$ that is consistent with all of the samples. Similarly to Section 3.1, one can show that $f^{(r+1)}$ achieves $O(\epsilon')$ error with respect to a large fraction of player distributions in each group. Consequently, the counters are increased, i.e., $\alpha_i^{(r+1)} > \alpha_i^{(r)}$, only for a small fraction of players. Finally, we show that with high probability, $\alpha_i^{(t)} \le 0.4t$ for any player $i \in [k]$, i.e., on each distribution $D_i$, at least $0.6t$ functions achieve error of at most $\epsilon'$.

The algorithm is formally described in Algorithm 2. The next theorem states our sample complexity upper bound for the centralized setting.

---

**Algorithm 2** CENTRALIZED LEARNING

---

$\alpha_i^{(0)} \leftarrow 0$ for each $i \in [k]$;
$t \leftarrow \left\lceil \frac{5}{2} \log_{8/7}(k) \right\rceil$; $\epsilon' \leftarrow \epsilon/6$;
$N_0^{(0)} \leftarrow [k]$; $N_c^{(0)} \leftarrow \emptyset$ for each $c \in [t]$;
**for** $r = 1, 2, \ldots, t$ **do**
    **for** $c = 0, 1, \ldots, t-1$ **do**
        **if** $N_c^{(r-1)} \ne \emptyset$ **then**
            Draw a sample set $S_c^{(r)}$ of size $m_{\epsilon'/16, \delta/(2t^2)}$ from $\widetilde{D}_c^{(r-1)} = \frac{1}{|N_c^{(r-1)}|} \sum_{i \in N_c^{(r-1)}} D_i$;
        **else** $S_c^{(r)} \leftarrow \emptyset$ ;
    **end**
    $f^{(r)} \leftarrow \mathcal{O}_\mathcal{F} \left( \bigcup_{c=0}^{t-1} S_c^{(r)} \right)$;
    $G_r \leftarrow \text{TEST}(f^{(r)}, [k], \epsilon', \delta/(2t))$;
    **for** $i = 1, \ldots, k$ **do** $\alpha_i^{(r)} \leftarrow \alpha_i^{(r-1)} + \mathbb{I}[i \notin G_r]$;
    **for** $c = 0, \ldots, t$ **do** $N_c^{(r)} \leftarrow \{i \in [k] : \alpha_i^{(r)} = c\}$;
**end**
**return** $\text{maj}(\{f^{(r)}\}_{r=1}^t)$;

---

**Theorem 3.2.** For any $\epsilon, \delta > 0$, and hypothesis class $\mathcal{F}$ of VC dimension $d$, Algorithm 2 $(\epsilon, \delta)$-learns $\mathcal{F}$ in the centralized collaborative PAC setting using $m$ samples, where

$$ m = O\left( \frac{\ln^2(k)}{\epsilon} \left( (d+k) \ln\left(\frac{1}{\epsilon}\right) + k \ln\left(\frac{1}{\delta}\right) \right) \right). $$

In particular, Algorithm 2 has $O(\ln^2(k))$ overhead when $k = O(d)$.

Turning to the theorem's proof, note that in Algorithm 2, $N_c^{(r-1)}$ represents the set of players for whom $c$ out of the $r-1$ functions learned so far have a large error, and $\widetilde{D}_c^{(r-1)}$ represents the mixture of distribution of players in $N_c^{(r-1)}$. Moreover, $G_r$ is the set of players for whom $f^{(r)}$ has a small error. The following lemma, whose proof appears in Appendix B.1, shows that with high probability each function $f^{(r)}$ has a small error on $\widetilde{D}_c^{(r-1)}$ for all $c$. Here and in the following, $t$ stands for $\left\lceil \frac{5}{2} \log_{8/7}(k) \right\rceil$ as in Algorithm 2.

**Lemma 3.3.** With probability $1 - \delta$, the following two properties hold for all $r \in [t]$:

1. For any $c \in \{0, \ldots, t-1\}$ such that $N_c^{(r-1)}$ is non-empty, $\text{err}_{\widetilde{D}_c^{(r-1)}}(f^{(r)}) \le \epsilon'/16$.
2. For any $i \in G_r$, $\text{err}_{D_i}(f^{(r)}) \le \epsilon'$, and for any $i \notin G_r$, $\text{err}_{D_i}(f^{(r)}) > \epsilon'/2$.

The next lemma gives an upper bound on $|N_c^{(r)}|$ — the number of players for whom $c$ out of the $r$ learned functions have a large error.

**Lemma 3.4.** With probability $1 - \delta$, for any $r, c \in \{0, \ldots, t\}$, we have $|N_c^{(r)}| \leq \binom{r}{c} \cdot \frac{k}{8^c}$.

*Proof.* Let $n_{r,c} = |N_c^{(r)}| = |\{i \in [k] : \alpha_i^{(r)} = c\}|$ be the number of players for whom $c$ functions in $f^{(1)}, \ldots, f^{(r)}$ do not have a small error. We note that $n_{0,0} = k$ and $n_{0,c} = 0$ for $c \in \{1, \ldots, t\}$. The next technical claim, whose proof appears in Appendix B.2, asserts that to prove this lemma, it is sufficient to show that for any $r \in \{1, \ldots, t\}$ and $c \in \{0, \ldots, t\}$, $n_{r,c} \leq n_{r-1,c} + \frac{1}{8} n_{r-1,c-1}$. Here we assume that $n_{r-1,-1} = 0$.

**Claim 3.5.** Suppose that $n_{0,0} = k$, $n_{0,c} = 0$ for $c \in \{1, \ldots, t\}$, and $n_{r,c} \leq n_{r-1,c} + \frac{1}{8} n_{r-1,c-1}$ holds for any $r \in \{1, \ldots, t\}$ and $c \in \{0, \ldots, t\}$. Then for any $r, c \in \{0, \ldots, t\}$, $n_{r,c} \leq \binom{r}{c} \cdot \frac{k}{8^c}$.

By definition of $\alpha_c^{(r)}$, $N_c^{(r)}$, and $n_{r,c}$, we have

$$n_{r,c} = \left|\{i \in [k] : \alpha_i^{(r)} = c\}\right| \leq \left|\{i \in [k] : \alpha_i^{(r-1)} = c\}\right| + \left|\{i \in [k] : \alpha_i^{(r-1)} = c - 1 \wedge i \notin G_r\}\right|$$

$$= n_{r-1,c} + \left|N_{c-1}^{(r-1)} \setminus G_r\right|.$$

It remains to show that $|N_{c-1}^{(r-1)} \setminus G_r| \leq \frac{1}{8} n_{r-1,c-1}$. Recall that $\widetilde{D}_{c-1}^{(r-1)}$ is the mixture of all distributions in $N_{c-1}^{(r-1)}$. By Lemma 3.3, with probability $1 - \delta$, $\mathrm{err}_{\widetilde{D}_{c-1}^{(r-1)}}(f^{(r)}) < \epsilon'/16$. Put another way, $\sum_{i \in N_{c-1}^{(r-1)}} \mathrm{err}_{D_i}(f^{(r)}) < \frac{\epsilon'}{16} \cdot |N_{c-1}^{(r-1)}|$. Thus, at most $\frac{1}{8} |N_{c-1}^{(r-1)}|$ players $i \in N_{c-1}^{(r-1)}$ can have $\mathrm{err}_{D_i}(f^{(r)}) > \epsilon'/2$. Moreover, by Lemma 3.3, for any $i \notin G_r$, we have that $\mathrm{err}_{D_i}(f^{(r)}) > \epsilon'/2$. Therefore,

$$\left|N_{c-1}^{(r-1)} \setminus G_r\right| \leq \left|\{i \in N_{c-1}^{(r-1)} : \mathrm{err}_{D_i}(f^{(r)}) > \epsilon'/2\}\right| \leq \frac{1}{8} \left|N_{c-1}^{(r-1)}\right| = \frac{1}{8} n_{r-1,c-1}.$$

This completes the proof. $\qquad\square$

We now prove Theorem 3.2 using Lemma 3.4.

*Proof of Theorem 3.2.* We first show that, with high probability, for any $i \in [k]$, at most $0.4t$ functions among $f^{(1)}, \ldots, f^{(t)}$ have error greater than $\epsilon'$, i.e., $\alpha_i^{(t)} < 0.4t$ for all $i \in [k]$. Note that by our choice of $t = \lceil \frac{5}{2} \log_{8/7}(k) \rceil$, we have $(8/7)^{0.4t} \geq k$. By Lemma 3.4 and an upper bound on binomial coefficients, with probability $1 - \delta$, for any integer $c \in [0.4t, t]$,

$$|N_c^{(t)}| \leq \binom{t}{c} \cdot \frac{k}{8^c} < \left(\frac{et}{c}\right)^c \cdot \frac{k}{8^c} < \frac{k}{(8/7)^c} \leq 1,$$

which implies that $N_c^{(t)} = \emptyset$. Therefore, with probability $1 - \delta$, $\alpha_i^{(t)} < 0.4t$ for all $i \in [k]$.

Next, we prove that $f = \mathrm{maj}(\{f^{(r)}\}_{r=1}^t)$ has error at most $\epsilon$ on every player distribution. Consider distribution $D_i$ of player $i$. By definition, $t - \alpha_i^{(t)}$ functions have error at most $\epsilon'$ on $D_i$. We refer to these functions as "good" functions. Note that for any instance $x$ that is mislabeled by $f$, at least $0.5t - \alpha_i^{(t)}$ good functions must make a wrong prediction. Therefore, $(t - \alpha_i^{(t)})\epsilon' \geq (0.5t - \alpha_i^{(t)}) \cdot \mathrm{err}_{D_i}(f)$. Moreover, with probability $1 - \delta$, $\alpha_i^{(t)} < 0.4t$ for all $i \in [k]$. Hence,

$$\mathrm{err}_{D_i}(f) \leq \frac{t - \alpha_i^{(t)}}{0.5t - \alpha_i^{(t)}} \epsilon' \leq \frac{0.6t}{0.1t} \epsilon' \leq \epsilon,$$

with probability $1 - \delta$. This proves that Algorithm 2 $(\epsilon, \delta)$-learns $\mathcal{F}$ in the centralized collaborative PAC setting.

Finally, we bound the sample complexity of Algorithm 2. Recall that $t = \Theta(\ln(k))$ and $\epsilon' = \epsilon/6$. At each iteration of Algorithm 2, we draw total of $t \cdot m_{\epsilon'/16,\delta/(4t^2)}$ samples from $t$ mixtures. Therefore, over $t$ time steps, we draw a total of

$$t^2 \cdot m_{\epsilon'/16,\delta/(4t^2)} = O\left(\frac{\ln^2(k)}{\epsilon} \cdot \left(d \ln\left(\frac{1}{\epsilon}\right) + \ln\left(\frac{1}{\delta}\right) + \ln\ln(k)\right)\right)$$

samples for learning $f^{(1)}, \ldots, f^{(t)}$. Moreover, the total number samples requested for subroutine $\text{TEST}(f^{(r)}, [k], \epsilon', \delta/(4t))$ for $r = 1 \ldots, t$ is

$$O\left(\frac{tk}{\epsilon} \cdot \ln\left(\frac{k}{\epsilon\delta}\right)\right) = O\left(\frac{\ln(k)}{\epsilon} \cdot \left(k \ln\left(\frac{1}{\epsilon}\right) + k \ln\left(\frac{1}{\delta}\right)\right) + \frac{\ln^2(k)}{\epsilon}k\right).$$

We conclude that the total sample complexity is

$$O\left(\frac{\ln^2(k)}{\epsilon}\left((d+k)\ln\left(\frac{1}{\epsilon}\right) + k \ln\left(\frac{1}{\delta}\right)\right)\right).$$

$\square$

We remark that Algorithm 2 is inspired by the classic boosting scheme. Indeed, an algorithm that is directly adapted from boosting attains a similar performance guarantee as in Theorem 3.2. The algorithm assigns a uniform weight to each player, and learns a classifier with $O(\epsilon)$ error on the mixture distribution. Then, depending on whether the function achieves an $O(\epsilon)$ error on each distribution, the algorithm updates the players' weights, and learns the next classifier from the weighted mixture of all distributions. An analysis similar to that of AdaBoost [9] shows that the majority vote of all the classifiers learned over $\Theta(\ln(k))$ iterations of the above procedure achieves a small error on *every* distribution. Similar to Algorithm 2, this algorithm achieves an $O(\ln^2(k))$ overhead for the centralized setting.

# 4 Sample Complexity Lower Bounds

In this section, we present lower bounds on the sample complexity of collaborative PAC learning. In Section 4.1, we show that any learning algorithm for the collaborative PAC setting incurs $\Omega(\log(k))$ overhead in terms of sample complexity. In Section 4.2, we consider the sample complexity required for obtaining *uniform convergence* across $\mathcal{F}$ in the collaborative PAC setting. We show that $\Omega(k)$ overhead is necessary to obtain such results.

## 4.1 Tight Lower Bound for the Personalized Setting

We now turn to establishing the $\Omega(\log(k))$ lower bound mentioned above. This lower bound implies the tightness of the $O(\log(k))$ overhead upper bound obtained by Theorem 3.1 in the personalized setting. Moreover, the $O(\log^2(k))$ overhead obtained by Theorem 3.2 in the centralized setting is nearly tight, up to a $\log(k)$ multiplicative factor. Formally, we prove the following theorem.

**Theorem 4.1.** For any $k \in \mathbb{N}$, $\epsilon, \delta \in (0, 0.1)$, and $(\epsilon, \delta)$-learning algorithm $\mathcal{A}$ in the collaborative PAC setting, there exist an instance with $k$ players, and a hypothesis class of VC-dimension $k$, on which $\mathcal{A}$ requires at least $3k \ln[9k/(10\delta)]/(20\epsilon)$ samples in expectation.

**Hard instance distribution.** We show that for any $k \in \mathbb{N}$ and $\epsilon, \delta \in (0, 0.1)$, there is a distribution $\mathcal{D}_{k,\epsilon}$ of "hard" instances, each with $k$ players and a hypothesis class with VC-dimension $k$, such that any $(\epsilon, \delta)$-learning algorithm $\mathcal{A}$ requires $\Omega(k \log(k)/\epsilon)$ samples in expectation on a random instance drawn from the distribution, even in the personalized setting. This directly implies Theorem 4.1, since $\mathcal{A}$ must take $\Omega(k \log(k)/\epsilon)$ samples on some instance in the support of $\mathcal{D}_{k,\epsilon}$. We define $\mathcal{D}_{k,\epsilon}$ as follows:

- Instance space: $\mathcal{X}_k = \{1, 2, \ldots, k, \perp\}$.
- Hypothesis class: $\mathcal{F}_k$ is the collection of all binary functions on $\mathcal{X}_k$ that map $\perp$ to 0.
- Target function: $f^*$ is chosen from $\mathcal{F}_k$ uniformly at random.
- Players' distributions: The distribution $D_i$ of player $i$ is either a degenerate distribution that assigns probability 1 to $\perp$, or a Bernoulli distribution on $\{i, \perp\}$ with $D_i(i) = 2\epsilon$ and $D_i(\perp) = 1 - 2\epsilon$. $D_i$ is chosen from these two distributions independently and uniformly at random.

Note that the VC-dimension of $\mathcal{F}_k$ is $k$. Moreover, on any instance in the support of $\mathcal{D}_{k,\epsilon}$, learning in the personalized setting is equivalent to learning in the centralized setting. This is due to the fact that given functions $f_1, f_2, \ldots, f_k$ for the personalized setting, where $f_i$ is the function assigned to player $i$, we can combine these functions into a single function $f \in \mathcal{F}_k$ for the centralized setting by defining $f(\perp) = 0$ and $f(i) = f_i(i)$ for all $i \in [k]$. Then, $\mathrm{err}_{D_i}(f) \le \mathrm{err}_{D_i}(f_i)$ for all $i \in [k]$.[1] Therefore, without loss of generality we focus below on the centralized setting.

**Lower bound for k = 1.** As a building block in our proof of Theorem 4.1, we establish a lower bound for the special case of $k = 1$. For brevity, let $\mathcal{D}_\epsilon$ denote the instance distribution $\mathcal{D}_{1,\epsilon}$. We say that $\mathcal{A}$ is an $(\epsilon, \delta)$-learning algorithm for the instance distribution $\mathcal{D}_\epsilon$ if and only if on any instance in the support of $\mathcal{D}_\epsilon$, with probability at least $1 - \delta$, $\mathcal{A}$ outputs a function $f$ with error below $\epsilon$. The following lemma, proved in Appendix C, states that any $(\epsilon, \delta)$-learning algorithm for $\mathcal{D}_\epsilon$ takes $\Omega(\log(1/\delta)/\epsilon)$ samples on a random instance drawn from $\mathcal{D}_\epsilon$.[2]

**Lemma 4.2.** For any $\epsilon, \delta \in (0, 0.1)$ and $(\epsilon, \delta)$-learning algorithm $\mathcal{A}$ for $\mathcal{D}_\epsilon$, $\mathcal{A}$ takes at least $\ln(1/\delta)/(6\epsilon)$ samples in expectation on a random instance drawn from $\mathcal{D}_\epsilon$. Here the expectation is taken over both the randomness in the samples and the randomness in drawing the instance from $\mathcal{D}_\epsilon$.

Now we prove Theorem 4.1 by Lemma 4.2 and a reduction from a random instance sampled from $\mathcal{D}_\epsilon$ to instances sampled from $\mathcal{D}_{k,\epsilon}$. Intuitively, a random instance drawn from $\mathcal{D}_{k,\epsilon}$ is equivalent to $k$ independent instances from $\mathcal{D}_\epsilon$. We show that any learning algorithm $\mathcal{A}$ that simultaneously solves $k$ tasks (i.e., an instance from $\mathcal{D}_{k,\epsilon}$) with probability $1 - \delta$ can be transformed into an algorithm $\mathcal{A}'$ that solves a single task (i.e., an instance from $\mathcal{D}_\epsilon$) with probability $1 - O(\delta/k)$. Moreover, the expected sample complexity of $\mathcal{A}'$ is only an $O(1/k)$ fraction of the complexity of $\mathcal{A}$. This transformation, together with Lemma 4.2, gives a lower bound on the sample complexity of $\mathcal{A}$.

*Proof Sketch of Theorem 4.1.* We construct an algorithm $\mathcal{A}'$ for the instance distribution $\mathcal{D}_\epsilon$ from an algorithm $\mathcal{A}$ that $(\epsilon, \delta)$-learns in the centralized setting. Recall that on an instance drawn from $\mathcal{D}_\epsilon$, $\mathcal{A}'$ has access to a distribution $D$, i.e., the single player's distribution.

- $\mathcal{A}'$ generates an instance $(\mathcal{F}_k, f^*, \{D_i\}_{i \in [k]})$ from the distribution $\mathcal{D}_{k,\epsilon}$ (specifically, $\mathcal{A}'$ knows the target function $f^*$ and the distributions), and then chooses $l \in [k]$ uniformly at random.
- $\mathcal{A}'$ simulates $\mathcal{A}$ on instance $(F_k, f^*, \{D_i\}_{i \in [k]})$, with $D_l$ replaced by the distribution $D$. Specifically, every time $\mathcal{A}$ draws a sample from $D_j$ for some $j \ne l$, $\mathcal{A}'$ samples $D_j$ and forwards the sample to $\mathcal{A}$. When $\mathcal{A}$ asks for a sample from $D_l$, $\mathcal{A}'$ samples the distribution $D$ instead and replies to $\mathcal{A}$ accordingly, i.e., $\mathcal{A}'$ returns $l$, together with the label, if the sample is 1 (recall that $\mathcal{X}_1 = \{1, \perp\}$), and returns $\perp$ otherwise.
- Finally, when $\mathcal{A}$ terminates and returns a function $f$ on $\mathcal{X}_k$, $\mathcal{A}'$ checks whether $\mathrm{err}_{D_j}(f) < \epsilon$ for every $j \ne l$. If so, $\mathcal{A}'$ returns the function $f'$ defined as $f'(1) = f(l)$ and $f'(\perp) = f(\perp)$. Otherwise, $\mathcal{A}'$ repeats the simulation process on a new instance drawn from $\mathcal{D}_{k,\epsilon}$.

Let $m_i$ be the expected number of samples drawn from the $i$-th distribution when $\mathcal{A}$ runs on an instance drawn from $\mathcal{D}_{k,\epsilon}$. We have the following two claims, whose proofs are relegated to Appendix C.

**Claim 4.3.** $\mathcal{A}'$ is an $(\epsilon, 10\delta/(9k))$-learning algorithm for $\mathcal{D}_\epsilon$.

**Claim 4.4.** $\mathcal{A}'$ takes at most $10/(9k) \sum_{i=1}^{k} m_i$ samples in expectation.

Applying Lemma 4.2 to $\mathcal{A}'$ gives $\sum_{i=1}^{k} m_i \ge \frac{3k \ln[9k/(10\delta)]}{20\epsilon}$, which proves Theorem 4.1. $\square$

## 4.2 Lower Bound for Uniform Convergence

We next examine the sample complexity required for obtaining *uniform convergence* across the hypothesis class $\mathcal{F}$ in the centralized collaborative PAC setting, and establish an overhead lower bound of $\Omega(k)$. Interestingly, our centralized learning algorithm (Algorithm 2) achieves $O(\log^2(k))$ overhead — it circumvents the lower bound by not relying on uniform convergence.

To be more formal, we first need to define uniform convergence in the cooperative PAC learning setting. We say that a hypothesis class $\mathcal{F}$ has the *uniform convergence property* with sample size $m_{\epsilon,\delta}^{(k)}$ if for any $k$ distributions $D_1, \ldots, D_k$, there exist integers $m_1, \ldots, m_k$ that sum up to $m_{\epsilon,\delta}^{(k)}$, such that when $m_i$ samples are drawn from $D_i$ for each $i \in [k]$, with probability $1 - \delta$, any function in $\mathcal{F}$ that is consistent with all the $m_{\epsilon,\delta}^{(k)}$ samples achieves error at most $\epsilon$ on every distribution $D_i$.

Note that the foregoing definition is a relatively weak adaptation of uniform convergence to the cooperative setting, as the integers $m_i$ are allowed to depend on the distributions $D_i$. But this observation only strengthens our lower bound, which holds despite the weak requirement.

**Theorem 4.5.** For any $k, d \in \mathbb{N}$ and $(\epsilon, \delta) \in (0, 0.1)$, there exists a hypothesis class $\mathcal{F}$ of VC-dimension $d$, such that $m_{\epsilon,\delta}^{(k)} \geq dk(1 - \delta)/(4\epsilon)$.

*Proof Sketch of Theorem 4.5.* Fix $k, d \in \mathbb{N}$ and $\epsilon, \delta \in (0, 0.1)$. We define instance $(\mathcal{F}, f^*, \{D_i\}_{i=1}^{k})$ as follows. The instance space is $\mathcal{X} = ([k] \times [d]) \cup \{\perp\}$, and the hypothesis class $\mathcal{F}$ contains all binary functions on $\mathcal{X}$ that map $\perp$ to 0 and take value 1 on at most $d$ points. The target function $f^*$ maps every element in $\mathcal{X}$ to 0. Finally, the distribution of each player $i \in [k]$ is given by $D_i((i, j)) = 2\epsilon/d$ for any $j \in [d]$ and $D_i(\perp) = 1 - 2\epsilon$.

Note that if a sample set contains strictly less than $d/2$ elements in $\{(i^*, 1), (i^*, 2), \ldots, (i^*, d)\}$ for some $i^*$, there is a consistent function in $\mathcal{F}$ with error strictly greater than $\epsilon$ on $D_{i^*}$, namely, the function that maps $(i, j)$ to 1 if and only if $i = i^*$ and $(i^*, j)$ is not in the sample set.

Therefore, to achieve uniform convergence, at least $d/2$ elements from $\mathcal{X} \setminus \{\perp\}$ must be drawn from each distribution. Since the probability that each sample is different from $\perp$ is $2\epsilon$, drawing $d/2$ such samples from $k$ distribution requires $\Omega(dk/\epsilon)$ samples. $\qquad\square$

A complete proof of Theorem 4.5 appears in Appendix C.

### Acknowledgments

We thank the anonymous reviewers for their helpful remarks and suggesting an alternative boosting-based approach for the centralized setting. This work was partially supported by the NSF grants CCF-1525971, CCF-1536967, CCF-1331175, IIS-1350598, IIS-1714140, CCF-1525932, and CCF-1733556, Office of Naval Research grants N00014-16-1-3075 and N00014-17-1-2428, a Sloan Research Fellowship, and a Microsoft Research Ph.D. fellowship. This work was done while Avrim Blum was working at Carnegie Mellon University.

## Footnotes

[1]In fact, when $f_i \in \mathcal{F}_k$, $\mathrm{err}_{D_i}(f) = \mathrm{err}_{D_i}(f_i)$ for all $i \in [k]$.

[2] Here we only assume that $\mathcal{A}$ is correct for instances in the support of $\mathcal{D}_\epsilon$, rather than being correct on *every* instance.

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
