[Supplementary Material]

# A Proof of Theorem 3.1

Let us first introduce a lemma that shows that the procedure TEST identifies the players who have a small error with respect to a given classifier. The proof of this lemma follows from the VC theorem (and Chernoff bound).

**Lemma A.1.** For any $f$, $N$, $\epsilon$, and $\delta$, with probability $1 - \delta$, $G = \text{TEST}(f, N, \epsilon, \delta)$ is such that

1. for any $i \in G$, $\text{err}_{D_i}(f) \le \epsilon$, and
2. for all $i \in N$, if $\text{err}_{D_i}(f) \le \frac{\epsilon}{2}$, then $i \in G$.

We now use the above lemma to prove Theorem 3.1.

By Lemma A.1, at every iteration of Algorithm 2, $G_r \leftarrow \text{TEST}(f^{(r)}, N_r, \epsilon, \delta')$ includes all players $i \in N_r$ for whom $\text{err}_{D_i}(f^{(r)}) \le \epsilon/2$ and no players $i \in N_r$ for whom $\text{err}_{D_i}(f^{(r)}) > \epsilon$. Therefore, it is sufficient to prove that for every $r = 1, \ldots, \lceil \log(k) \rceil$, $|G_r| \ge \frac{1}{2}|N_r|$, in which case the algorithm ends after $\lceil \log(k) \rceil$ iterations and every player has received a function with error at most $\epsilon$ on his distribution.

In the remainder of the proof, we show that $|G_r| \ge \frac{1}{2}|N_r|$ for any $r$ with probability $1 - \delta/\log(k)$. Recall that $f^{(r)}$ is learned by taking a sample of size $m_{\epsilon/4, \delta'}$ over distribution $\tilde{D}_r = \frac{1}{|N_r|} \sum_{i \in N_r} D_i$. Therefore, with probability $1 - \delta'$, $\text{err}_{\tilde{D}_r}(f^{(r)}) \le \epsilon/4$. By Markov's inequality, with probability $1 - \delta'$, $f^{(r)}$ has an error of $\epsilon/2$ for at least half of the players in $N_r$. Using Lemma A.1, with probability at most $\delta'$, one or more such players are not included in $G_r$. Therefore, with overall probability $1 - 2\delta' = 1 - \delta/\log(k)$, $|G_r| \ge |N_r|/2$.

As for the sample complexity, TEST is called $\log(k)$ times, and requests

$$O\left(\frac{k \log(k)}{\epsilon} \cdot \ln\left(\frac{k}{\epsilon\delta}\right)\right) = O\left(\frac{\log(k)}{\epsilon}\left(k \ln\left(\frac{1}{\epsilon}\right) + k \ln\left(\frac{k}{\delta}\right)\right)\right)$$

samples overall. Moreover, we learn a total of $\log(k)$ classifiers $f^{(r)}$, requesting

$$\log(k) \cdot m_{\epsilon/4, \delta'} = O\left(\frac{\log(k)}{\epsilon}\left(d \ln\left(\frac{1}{\epsilon}\right) + \ln\left(\frac{\log(k)}{\delta}\right)\right)\right).$$

samples overall. The sample complexity follows from these two bounds. □

# B Omitted Proofs from Section 3.2

## B.1 Proof of Lemma 3.3

Recall that for any $r, c$ such that $N_c^{(r-1)}$ is non-empty, $f^{(r)}$ is consistent with the $m_{\epsilon'/16, \delta/(2t^2)}$ samples drawn from $\widetilde{D}_c^{(r-1)}$. By the VC theorem, $\text{err}_{\widetilde{D}_c^{(r-1)}}(f^{(r)}) \le \epsilon'/16$ holds with probability at least $1 - \delta/(2t^2)$. Also, by Lemma A.1, the second statement holds with probability $1 - \delta/(2t)$ for each $r \in [t]$. It follows from the union bound that with probability at least

$$1 - t^2 \cdot \delta/(2t^2) - t \cdot \delta/(2t) = 1 - \delta,$$

the two statements holds for any $r \in [t]$ simultaneously.

## B.2 Proof of Claim 3.5

We prove the claim by induction on $r$. Since $n_{0,0} = k$ and $n_{0,c} = 0$ for any $c \in [t]$, the inequality holds for $r = 0$. Suppose that for some $r \in [t]$, the inequality

$$n_{r',c} \le \binom{r'}{c} \cdot \frac{k}{8^c}$$

holds for $r' = r - 1$ and any $c \in \{0, 1, \ldots, t\}$. Then we have for any $c \in \{0, 1, \ldots, t\}$,

$$n_{r,c} \le n_{r-1,c} + n_{r-1,c-1}/8 \le \binom{r-1}{c} \cdot \frac{k}{8^c} + \binom{r-1}{c-1} \cdot \frac{k}{8^{c-1} \times 8} = \binom{r}{c} \cdot \frac{k}{8^c}.$$

Therefore, we conclude that the inequality holds for any $r, c \in \{0, 1, \ldots, t\}$.

# C    Omitted Lower Bound Proofs

## C.1    Proof of Lemma 4.2

Recall that in any instance in the support of $\mathcal{D}_\epsilon$, the instance space is $\mathcal{X} = \{1, \perp\}$, while the hypothesis class is $\mathcal{F} = \{f_0, f_1\}$, where $f_i(\perp) = 0$ and $f_i(1) = i$ for $i \in \{0, 1\}$.

Fix an $(\epsilon, \delta)$-learning algorithm $\mathcal{A}$ for distribution $\mathcal{D}_\epsilon$. Suppose $\mathcal{A}$ runs on an instance drawn from $\mathcal{D}_\epsilon$ with a degenerate distribution $D$. For $i \in \{0, 1\}$, let $\mathcal{E}_i$ denote the event that $\mathcal{A}$ outputs $f_i$. Define random variable $T$ as the number of samples drawn by $\mathcal{A}$ before it terminates, and let $p_n = \Pr[T = n | \mathcal{E}_0]$.

Now we consider the situation that $\mathcal{A}$ runs on the instance with $D'(1) = 2\epsilon$, $D'(\perp) = 1 - 2\epsilon$, and the target function is $f_1$. On this instance, with probability at least $p_n \cdot (1 - 2\epsilon)^n$, $\mathcal{A}$ outputs $f_0$ after drawing exactly $n$ samples, all of which are $\perp$. Since $D'(1) = 2\epsilon$ and the target function is $f_1$, $\text{err}_{D'}(f_0) = 2\epsilon > \epsilon$, i.e., $\mathcal{A}$ outputs a function with error greater than $\epsilon$ on $D'$. Since $\mathcal{A}$ is an $(\epsilon, \delta)$-learning algorithm for $\mathcal{D}_\epsilon$,

$$\delta \geq \sum_{n=0}^{\infty} p_n \cdot (1 - 2\epsilon)^n = \mathbb{E}\left[(1 - 2\epsilon)^T \big| \mathcal{E}_0\right].$$

By Jensen's inequality and the convexity of the function $\log_{1-2\epsilon} x$ for $\epsilon \in (0, 0.1)$, we have

$$\mathbb{E}\left[T | \mathcal{E}_0\right] \geq \log_{1-2\epsilon} \mathbb{E}\left[(1 - 2\epsilon)^T | \mathcal{E}_0\right] \geq \log_{1-2\epsilon} \delta = \frac{\ln(1/\delta)}{\ln[1/(1 - 2\epsilon)]} \geq \frac{\ln(1/\delta)}{3\epsilon}.$$

Here the last step holds since $\ln[1/(1 - 2\epsilon)] \leq 3\epsilon$ for any $\epsilon \in (0, 0.1)$. A similar argument (using the same distribution $D'$, but the target function $f_0$ instead of $f_1$) gives $\mathbb{E}\left[T | \mathcal{E}_1\right] \geq \ln(1/\delta)/(3\epsilon)$.

Therefore, $\mathcal{A}$ takes at least $\ln(1/\delta)/(3\epsilon)$ samples in expectation when the distribution $D$ is degenerate, which happens with probability $1/2$ for an instance drawn from $\mathcal{D}_\epsilon$. Therefore, the expected sample complexity of $\mathcal{A}$ on a random instance sampled from $\mathcal{D}_\epsilon$ is lower bounded by

$$\frac{1}{2} \cdot \frac{\ln(1/\delta)}{3\epsilon} = \frac{\ln(1/\delta)}{6\epsilon}.$$

$\square$

## C.2    Claims in the Proof of Theorem 4.1

*Proof of Claim 4.3.* Let $p_i$ be the probability that, on a random instance drawn from $\mathcal{D}_{k,\epsilon}$, the function $f$ returned by $\mathcal{A}$ satisfies $\text{err}_{D_i}(f) > \epsilon$ and $\text{err}_{D_j}(f) \leq \epsilon$ for any $j \neq i$. By assumption, $\sum_{i=1}^{k} p_i \leq \delta$.

Let random variable $T$ denote the number of times that $\mathcal{A}'$ repeats the simulation process (it repeats the process every time the condition $\forall j \neq l, \ \text{err}_{D_j}(f) < \epsilon$ is violated). Let $\mathcal{E}_i$ denote the event that $\mathcal{A}'$ returns a function with error greater than $\epsilon$ and $T = i$. Clearly, $\mathcal{E}_i$ implies:

1. The simulated algorithm $\mathcal{A}$ fails to return a function with an error smaller than $\epsilon$ on every distribution in each of the first $i - 1$ simulations, which happens with probability at most $\delta^{i-1}$.
2. In the $i$-th iteration, $\mathcal{A}$ returns a function $f$ such that $\text{err}_{D_j}(f) \leq \epsilon$ for $j \neq l$, yet $\text{err}_{D_l}(f) > \epsilon$. This happens with probability $p_l$.

Recall that $l$ is drawn uniformly at random from $[k]$. Thus,

$$\Pr[\mathcal{E}_i] \leq \delta^{i-1} \cdot \frac{1}{k} \sum_{i=1}^{k} p_i \leq \delta^i/k.$$

Overall, the probability that $\mathcal{A}'$ returns a function with error greater than $\epsilon$ is bounded by

$$\sum_{i=1}^{\infty} \Pr[\mathcal{E}_i] \leq \sum_{i=1}^{\infty} \delta^i/k = \frac{\delta}{k(1 - \delta)} \leq \frac{10\delta}{9k}.$$

which proves that $\mathcal{A}'$ is an $(\epsilon, 10\delta/(9k))$-learning algorithm for $\mathcal{D}_\epsilon$.

$\square$

*Proof of Claim 4.4.* Let random variable $T$ denote the number of times that $\mathcal{A}'$ repeats the simulation process. Let $X_1, X_2, \dots$ be the number of samples drawn from distribution $D$ in each simulation. Note that these random variables are independently and identically distributed, so by Wald's equation,

$$\mathbb{E}\left[\sum_{i=1}^{T} X_i\right] = \mathbb{E}\left[T\right] \cdot \mathbb{E}\left[X_1\right].$$

For any positve integer $i$, $T \geq i$ holds only if the simulated algorithm $\mathcal{A}$ fails to return a function with an error smaller than $\epsilon$ on every distribution in each of the first $i-1$ simulations. By assumption, this happens with probability at most $\delta^{i-1}$. Therefore,

$$\mathbb{E}\left[T\right] = \sum_{i=1}^{\infty} \Pr[T \geq i] \leq \sum_{i=0}^{\infty} \delta^i \leq \frac{1}{1-\delta} \leq \frac{10}{9}.$$

Note that conditioning on the value of $l$ in the first iteration of $\mathcal{A}'$, $\mathcal{A}'$ draws $m_l$ samples from $D$ in expectation. Since $l$ is uniformly distributed in $[k]$,

$$\mathbb{E}\left[X_1\right] = \frac{1}{k} \sum_{i=1}^{k} m_i,$$

and the expected number of samples taken by $\mathcal{A}'$ in total is at most

$$\mathbb{E}\left[T\right] \cdot \mathbb{E}\left[X_1\right] \leq \frac{10}{9k} \sum_{i=1}^{k} m_i.$$

$\square$

### C.3   Proof of Theorem 4.5

Fix $k, d \in \mathbb{N}$ and $\epsilon, \delta \in (0, 0.1)$. We define instance $(\mathcal{F}, f^*, \{D_i\}_{i=1}^k)$ as follows:

- Instance space: $\mathcal{X} = ([k] \times [d]) \cup \{\perp\}$.
- Hypothesis class: $\mathcal{F}$ is the collection of all binary functions on $\mathcal{X}$ that map $\perp$ to 0 and take value 1 on at most $d$ points.
- Target function: $f^*$ maps every element in $\mathcal{X}$ to 0.
- Players' distributions: for each player $i \in [k]$, $D_i((i,j)) = 2\epsilon/d$ for any $j \in [d]$ and $D_i(\perp) = 1 - 2\epsilon$.

Let $m_1, m_2, \dots, m_k$ be integers such that $m_1 + m_2 + \cdots + m_k = m_{\epsilon,\delta}^{(k)}$, and when $m_i$ samples are drawn from $D_i$ for each $i \in [k]$, with probability $1 - \delta$, any consistent function in $\mathcal{F}$ has an error at most $\epsilon$ on every $D_i$. We consider the following algorithm $\mathcal{A}$ that proceeds in rounds: in each round, $\mathcal{A}$ draws $m_i$ samples from $D_i$ for each $i \in [k]$. $\mathcal{A}$ terminates if at the end of some round, $\text{err}_{D_i}(f) \leq \epsilon$ for all $i \in [k]$ and any function $f \in \mathcal{F}$ that is consistent with the $m_{\epsilon,\delta}^{(k)}$ samples. In expectation, $\mathcal{A}$ terminates after at most $1/(1-\delta)$ rounds, and takes at most $m_{\epsilon,\delta}^{(k)}/(1-\delta)$ samples.

Note that if a sample set contains strictly less than $d/2$ elements in $\{(i^*,1),(i^*,2),\dots,(i^*,d)\}$ for some $i^*$, there is a consistent function in $\mathcal{F}$ with error strictly above $\epsilon$ on $D_{i^*}$, namely, the function that maps $(i,j)$ to 1 if and only if $i = i^*$ and $(i^*,j)$ is not in the sample set. Therefore, when $\mathcal{A}$ terminates, at least $d/2$ elements from $\mathcal{X} \setminus \{\perp\}$ have been drawn from each distribution.

Note that the probability that each sample is different from $\perp$ is $2\epsilon$, so, in expectation, $(d/2) \cdot (1/(2\epsilon)) = d/(4\epsilon)$ samples from each distribution are required to draw $d/2$ samples from $\mathcal{X} \setminus \{\perp\}$. Therefore, we have $m_{\epsilon,\delta}^{(k)}/(1-\delta) \geq dk/(4\epsilon)$, which proves the theorem. $\square$

## D   Extension to the Non-realizable Setting

In this section, we generalize our sample complexity upper bounds in Section 3 to the *non-realizable setting*, where we have a weaker assumption on the consistency between players' distributions. Instead of assuming a perfect target function in $\mathcal{F}$ with zero error on every distribution, we consider the case that there exists $f^* \in \mathcal{F}$ with $\text{err}_{D_i}(f^*) \leq \epsilon/100$ for all $i \in [k]$. Our goal is still to output

a single function or multiple functions, such that the function assigned to each player has an error below $\epsilon$ on that player's distribution. We call this the *non-realizable collaborative PAC setting*, and prove analogues of Theorems 3.1 and 3.2.

**Theorem D.1.** There is an $(\epsilon, \delta)$-learning algorithm in the non-realizable personalized collaborative PAC setting using $m$ samples, where

$$m = O\left(\frac{\log(k)}{\epsilon}\left((d+k)\log\left(\frac{1}{\epsilon}\right) + k\log\left(\frac{k}{\delta}\right)\right)\right).$$

**Theorem D.2.** There is an $(\epsilon, \delta)$-learning algorithm in the non-realizable centralized collaborative PAC setting using $m$ samples, where

$$m = O\left(\frac{\log^2(k)}{\epsilon}\left((d+k)\ln\left(\frac{1}{\epsilon}\right) + k\ln\left(\frac{1}{\delta}\right)\right)\right).$$

We prove Theorem D.2 by slightly adapting Algorithm 2; Theorem D.1 can be proved similarly.

*Proof of Theorem D.2.* Recall that in each round $r$ of Algorithm 2, we draw

$$m_{\epsilon'/16, \delta/(2t^2)} = m_{\epsilon/96, \delta/(2t^2)}$$

samples from each mixture $\widetilde{D}_c^{(r-1)}$ and query the oracle $\mathcal{O}_\mathcal{F}$ on the union of all the samples. Now suppose that, instead, we draw from $\widetilde{D}_c^{(r-1)}$ a dataset $S_c^{(r)}$ of size

$$C \cdot \frac{d\ln(1/\epsilon) + \ln(2t^2/\delta)}{\epsilon}.$$

By [1, Theorem 5.7], we can choose a sufficiently large constant $C$ such that with probability $1 - \delta/(2t^2)$, for any function $f \in \mathcal{F}$:

1. $\text{err}_{\widetilde{D}_c^{(r-1)}}(f) \le \epsilon/100$ implies $\text{err}_{S_c^{(r)}}(f) \le \epsilon/98$.
2. $\text{err}_{\widetilde{D}_c^{(r-1)}}(f) > \epsilon/96$ implies $\text{err}_{S_c^{(r)}}(f) > \epsilon/98$.

Then we choose $f^{(r)}$ such that the empirical error of $f^{(r)}$ on every dataset $S_c^{(r)}$ is upper bounded by $\epsilon/98$. Note that given that the two conditions above hold, such a function $f^{(r)}$ always exists, since we assume that $\text{err}_{\widetilde{D}_c^{(r-1)}}(f^*) \le \epsilon/100$. Other parts of Algorithm 2 remain unchanged.

The modified algorithm indeed $(\epsilon, \delta)$-learns in the non-realizable collaborative PAC setting. Since we guarantee that with probability $1 - \delta/(2t^2)$, the error of $f^{(r)}$ on $\widetilde{D}_c^{(r)}$ is bounded by $\epsilon/96$, Lemma 3.3 and the rest of the proof still holds. Furthermore, the (asymptotic) sample complexity of the algorithm does not change, as the number of samples drawn from each mixture only increases by a constant factor. □