[Reviews · NeurIPS 2017]

Reviewer 1



**Edit** I have read the other reviews and rebuttal. I thank the authors for their feedback and clarification and my positive evaluation of this paper remains unchanged. The authors give theoretical guarantees for collaborative learning with shared information under the PAC regime. They show that the sample complexity for (\epsilon,\delta)-PAC learning k classifiers for a shared problem (or a size k majority-voting ensemble) only needs to grow by a factor of approximately 1+log(k) (or 1+log^2(k)) rather than a factor of k as naive theory might predict. The authors provide pseudocode for two algorithms treated by the theory. These were correct as far as I could see, though I didn't implement either. The paper is mostly clearly written, and strikes a sensible balance between what is included here and what is in the supplementary material, and I enjoyed reading it. I checked the paper and sections A and B of the supplementary material quite carefully, and skimmed C and D. There were no obvious major errors I could find. It would, of course, have been ideal to see experimental results corroborating the theory, but space is limited and the theory is the main contribution. Hopefully those will appear in a later extended version. I have mostly minor comments: Somewhere it should be highlighted explicitly that t is a placeholder for log_{8/7}(k) thus although t is indeed logarithmic in k it is also not *very* small compared to k unless k is large (few hundred say) and in this setting a very large k strikes me as unlikely. On the other hand some central steps of the proof involve union bounding over 2t terms, which is usually disastrous, but here the individual sample complexities for each classifier still only grow like log(2t/\delta), i.e. loglog(k) and you may like to highlight that as well. I think it is also worth mentioning that the guarantees for the second algorithm depend on a large enough quorum of individual voters being right at the same time (w.h.p) thus the guarantee I think is for the whole ensemble and not for parts of it i.e. since we don't know which individual classifiers are right and which are not. In particular, in practice could early stopping or communications errors potentially invalidate the theoretical guarantees here? Line 202 and elsewhere use the notation of line 200, i.e. n_{r-1,c} + \frac{1}{8}n_{r-1,c-1}, since it is much clearer. Line 251 can you please clarify the notation? \mathcal{D}_{\orth} is a distribution over - what? - with relation to the \mathcal{D}_i? It seems like \mathcal{F}_k and \mathcal{D}_k have some dependency on one another? Line 393 in supplementary should be \forall r < r' \in [t]. A comment - the theory for the second algorithm looks like it could be adapted to, say, a bagged ensemble with only very minor modifications. There is little in the way of good theory for ensemble classification, in particular when is an ensemble of "weak" classifiers better than a single "strong" classifier is not well understood.

Reviewer 2



SUMMARY This paper considers a generalization of the PAC learning model in which there are k learners, each receiving i.i.d. data from a different distribution D_i but labeled by a common target function f*. The goal is either: -- (personalized) obtain k functions f_1, ..., f_k that do well on distributions D_1, ..., D_k respectively or -- (centralized) obtain one function f that does well on all k distributions. It is assumed that we are in the "realizable" case, i.e. a class F is known that contains f*. Suppose that for learning an epsilon-good function, the normal PAC bound (for just one learner) is m. The naive approach to solving either of the two generalized problems (again, with error at most epsilon) would result in a total of O(km) samples being drawn, basically m per learner. Instead, the authors present two simple and fairly clever schemes that solve the personalized and centralized problems with a total of just O(m polylog(k)) samples. COMMENTS This is a very well written paper that gives lots of intuition about the problem and the proof techniques. The specific problems considered here have some of the flavor of domain adaptation, and should be of fairly broad interest, at least at a conceptual level. Still, it would have been nice if the authors had given some more compelling real-world examples as motivation, some for the personalized case and some for the centralized case.

Reviewer 3



This paper studies the problem in which k players have examples labeled by the same function but with points wrawn from k different distributions. The goal is to use these examples to (a) obtain a classifier that has low error on all k distributions (b) obtain a personalized classifier for each of the players that has low error. The paper shows simple algorithms (or more accurately reductions to the usual PAC learning) that show how to achieve this with log^2 k and log k factor overheads respectively. This improves the naive implementation that would require factor k more samples. The problem with just combining the samples is that the player that has a harder distribution might get all the errors. This work shows that log k factor overhead is unavoidable and also that achieving this upper bound requires an algorithm that uses a larger hypothesis class than that used to generate the labels. There is also an extension to non-realizable setting althougt only for tiny noise rate that allows the same analysis to go through. The problem setting is fairly natural and has been studied before (e.g. Balcan,Blum,Fine,Mansour 2012 although in the context of reducing communication rather than ensuring the largest error). The question is also fairly natural and this work gives some basic insights into it. The slightly more involved centralized case uses a boosting like algorithm. It is unclear why the authors did not directly use a reweighting scheme from boosting (unless I'm missing something this would be both simpler and give log k instead of log^2 k dependence). Overall the paper studies a clean theoretical model of collaborative learning and gives simple insights related to ensuring that all players have a maximum error guarantee. A weaker side of the paper is that the solutions are entirely straightforward given the problem and there is no treatment of the potentially more interesting non-realizable setting. >> Thanks for the clarification about the log factors when using boosting.